# Research on the Improvement of Concrete Autogenous Self-healing Based on the Regulation of Cement Particle Size Distribution (PSD)

**DOI:** 10.3390/ma12172818

**Published:** 2019-09-02

**Authors:** Lianwang Yuan, Shuaishuai Chen, Shoude Wang, Yongbo Huang, Qingkuan Yang, Shuai Liu, Jinbang Wang, Peng Du, Xin Cheng, Zonghui Zhou

**Affiliations:** 1Shandong Provincial Key Laboratory of Preparation and Measurement of Building Materials, Jinan 250022, Shandong, China (L.Y.) (S.C.) (S.W.) (Y.H.) (Q.Y.) (S.L.) (J.W.) (P.D.) (X.C.); 2School of Materials Science and Engineering, University of Jinan, Jinan 250022, Shandong, China

**Keywords:** PSD of cement, autogenous self-healing, ultrasonic test, continued hydration, CaCO_3_ precipitation

## Abstract

Overgrinding of Portland cement brings excessive shrinkage and poor self-healing ability to concrete. In this paper, through the ultrasonic test and optical micrograph observation, the self-healing properties of concrete prepared by cement with different particle size distributions were studied. Besides, the effect of carbonation and continued hydration on self-healing of concrete was analyzed. Results show that, for the Portland cement containing more particles with the size 30~60 μm, the concrete could achieve a better self-healing ability of concrete at 28 days. For the two methods to characterize the self-healing properties of concrete, the ultrasonic test is more accurate in characterizing the self-healing of internal crack than optical micrograph observation. The autogenous self-healing of concrete is jointly affected by the continued hydration and carbonation. At 7 days and 30 days, the autogenous self-healing of concrete is mainly controlled by the continued hydration and carbonation, respectively. The cement particle size could affect the continued hydration by affecting un-hydrated cement content and the carbonation by affecting the Ca(OH)_2_ content. Therefore, a proper distribution of cement particle size, which brings a suitable amount of Ca(OH)_2_ and un-hydrated cement, could improve the self-healing ability of concrete.

## 1. Introduction

Concrete cracks provide the access for harmful substances, containing chloride, CO_2_ and sulfate ions, to entering the interior structure. It accelerates the corrosion of steel bar and the concrete failure. Therefore, it is necessary to repair the cracks to extend the service life of concrete structures. The manual repair requires much manpower and material. But it can only heal the accessible surface cracks [1]. It is difficult to find the micro-cracks in time [2], which will continue to grow and expand. Therefore, concrete with self-healing ability is desired to repair both the surface micro-crack and internal micro-crack.

In general, cement-based material has the self-healing ability of the concrete matrix itself, which is usually called as the autogenous self-healing. The autogenous self-healing ability enables the micro-cracks of the concrete to be healed. It is different from the self-sealing by the restoration of strength of concrete [3,4]. The limitation of crack width and the carbonation benefits autogenous self-healing [5]. When the micro-cracks occur and the liquid water is available, the continued hydration [6] and carbonation make the concrete exhibiting the self-healing ability [1]. Furthermore, the impurities in the water and the swelling of matrix are also beneficial to seal the micro-cracks [7]. However, the above mentioned autogenous self-healing ability of concrete is poor because of the overgrinding of cement, which brings less un-hydrated cement [8].

To obtain an excellent self-healing ability, many researchers have developed some new additive materials for concrete. Jefferson et al. and Teal et al. found that the shape memory materials (SMM) can remember the original shape and have a tendency to recovery. Therefore, the addition of SMM to concrete could repair the cracks of concrete [9,10]. Polymer repair agents and bacteria can also improve the self-healing ability of concrete cracks [11,12]. However, these materials need to be safely embedded in the concrete by the protection of microcapsules and hollow tubes. Wang et al. and Li et al. have found that capsule-coated repair agents could repair the cracks and improve the mechanical properties [13,14]. In addition, the hollow tubes [15,16] and vascular networks [17] can also be used to store and deliver the repair agents in the concrete, which can provide the repeated healing by the supply of external repair agents. Belie et al. found that the bacteria produce precipitation by mineralization to repair the cracks when microcapsules are ruptured [18,19]. To guarantee the survival of bacteria, it is also necessary to add nutrients into the concrete by microcapsules [20,21]. In addition, Ferrara et al., De Nardi et al., and Sisomphon et al. have found that the expansive additives and crystalline admixtures can also improve the self-healing ability of concrete [4,22,23]. Through the addition of the above mentioned repair materials to concrete, the autonomous self-healing ability is improved and the cracks can be repaired. However, the high cost limits its application.

Therefore, many researchers are focused on the investigation of repair agents with low cost. From the perspective of economy, the cementitious materials were studied firstly to improve the autogenous self-healing of concrete. Rahmani et al. used the coarse cement particles as repair agents to study the self-healing of concrete and found the compressive strength and permeability improved after self-healing [24]. Zhou et al. also confirmed an improved self-healing ability by adding coarse cement particles into concrete [25]. At present, to promote the early strength development of the concrete, the overgrinding of cement particles brings about the excessive energy consumption, poor durability of concrete and poor self-healing ability [26,27,28]. Therefore, to improve the self-healing ability of concrete, the PSD of Portland cement should be optimized.

When the micro-cracks produced in concrete structures surface, the optical micrograph measurement is the most intuitive method to monitor. As for the internal cracks, the ultrasonic nondestructive tests are the ideal methods to evaluate. The ultrasonic pulse velocity (UPV) test can reflect the damage of concrete and also be used to characterize the self-healing of cracks. Concrete damaged by freeze-thaw resulted in a reduced UPV to 78~45% of undamaged values, and the self-healing for cracked specimens gave recovery in UPV of 50~100% [29,30]. Ultrasonic waveforms and frequencies can also evaluate the self-healing properties of concrete based on the high repeatability of the energy transmission in a stable state [31].

In this paper, the influence of cement particle size on the concrete self-healing was studied, aiming to optimize the PSD of cement to improve the self-healing ability of concrete. Optical microscope observation was used to measure the size of crack closure. UPV was used to evaluate the self-healing of internal cracks. Furthermore, the ultrasonic waveform and frequency tests were also proposed to evaluate the self-healing ability of concrete more accurately. The PSD of cement was optimized by the relationship between self-healing ability and PSD of cement. Besides, the effect of carbonation and continued hydration on the self-healing ability of concrete was also investigated.

## 2. Materials and Methods

### 2.1. Raw Materials

The Portland cement clinker (Shandong Lubi Company, Jinan, China) and the natural gypsum (CaSO_4_·2H_2_O) were used as the cementitious materials. The chemical compositions of the clinker and gypsum are shown in Table 1. Their chemical compositions are consistent with ordinary Portland cement for commercial application. The f-CaO content of clinker is less than 0.5% wt. The natural quartz standard sand (Xiamen ISO Company, Xiamen, China) was used as the aggregate. The content of SiO_2_ is more than 96% and the impurity content is less than 0.2%. The standard sand with continuous grading contains the coarse sand (1.0~2.0 mm) of 33.3%, the medium sand (0.5~1.0 mm) of 33.3% and the fine sand (0.08~0.5 mm) of 33.3%. Superplasticizer (polycarboxylate, Shandong Academy of Building Research, Jinan, China) with a water reducing ratio of 32% was used to improve the workability of the fresh mortar. 

### 2.2. Preparation Procedures

#### 2.2.1. Cement Preparation

The mixtures of clinker and gypsum with a mass ratio of 19:1 were ground into varying sizes of cement particles by the ball milling. The PSD of cement is shown in Table 2. The PSD of ordinary Portland cement (OPC, Grading 42.5) is also shown in Table 2 as a comparison. The PSD of all groups are coarser than the OPC, because of the overgrinding of current cement and the poor self-healing ability.

#### 2.2.2. Sample Preparation

The cubes of cement paste (20 × 20 × 20 mm) and the prisms of mortar (40 × 40 × 160 mm) were cast in accordance with EN 1015-11 [32]. The ratio of water to cement is 0.4 and the ratio of binder to sand is 1:3. The cementitious content was used in the mixture design is 550 kg/m^3^. Then, the cement and water were mixed in a laboratory mixer (Changsha Deke Instrument Equipment Co., Ltd., Changsha, China) and placed in 20 × 20 × 20 mm molds and the mortar mixtures were placed in 40 × 40 × 160 mm molds. The cement paste and mortar were compacted using the vibrating table. The cube and prism molds were covered with a plastic foil and cured 24 h at 25 °C and 95% relative humidity. After demolding, all the samples were cured in relative humidity (RH) of 95%, 25 °C for 28 days. Each group has three prisms (40 × 40 × 160 mm) and six cubes (20 × 20 × 20 mm) for test to ensure the repeatability and reliability, and sufficient self-healing products for analysis.

#### 2.2.3. Prefabricating Cracks and Self-healing Evaluation

The micro-cracks in the cement paste were prepared by a method referred to as splitting experiments [33,34], while the micro-cracks in the mortar were prepared by a three-point bending method [1,35]. In order to benefit the initial crack generation, a notch (5 mm deep and 2 mm wide) in the middle of each mortar sample (40 × 40 × 160 mm) bottom was sawn using a circular saw, instead of being precast [36]. A steel bar with a diameter of 2 mm was used for the splitting test. All processes were implemented with a MTS servo-hydraulic testing machine (MTS (China) corporation, Shanghai, China). Displacement-controlled test mechanism was carried out at a rate of 0.2 mm/min to get an ideal micro-crack of each sample. The schematics are shown in Figure 1a,b, respectively. After pre-cracked, the samples were all fixed with epoxy resin to keep the micro-crack width independent of external forces. Then, all samples were placed in water for self-healing. The period of 30 d for self-healing was selected when the first completely closed crack of these groups was found.

### 2.3. Evaluation of Self-healing 

The non-destructive tests were used to determine the self-healing of concrete cracks, which include the optical micrograph measurement, UPV, ultrasonic waveform and frequency analysis.

#### 2.3.1. Optical Micrograph Measurement of Micro-crack Closure

The changes of surface micro-crack width can be used to evaluate the self-healing of concrete directly. The micro-crack width was measured at a specific magnification using a Crack Width Gauge (Shenzhen Siwell Company, Shenzhen, China). An enlarged real-time crack image would appear on the LCD screen when the probe was placed on the micro-crack and opened. The micro-crack width was analyzed by the specialized measurement software with an accuracy of 0.1 μm. Three different positions for each micro-crack were measured and marked. The average micro-crack width was calculated and used to indicate the width of this micro-crack. After self-healing, the micro-cracks were measured again at the marked positions.

#### 2.3.2. UPV

UPV test of mortar samples was measured by a TICO concrete ultrasonic test device (Shenborui Company, Shenzhen, China). The parameters were set as a pulse voltage of 1 kV and the resolution of 0.1 μs. Two sensors with a natural frequency of 54 KHz were used to transmit and receive signals, respectively. The two sensors were placed on the same side of the prisms. The distance between two sensors was 97 mm. Figure 2a shows the schematic diagram of UPV test.

#### 2.3.3. Ultrasonic Waveform and Frequency

The ultrasonic wave was generated and transmitted by a Tektronix AFG Model 3022C arbitrary function generator (Tektronix (China) Co., Ltd., Shanghai, China), while it received and displayed by a Tektronix MDO 3024 Mixed Domain Oscilloscope (Tektronix (China) Co., Ltd., Shanghai, China). The parameters were set as the ultrasonic frequency of 107 kHz and the high and low levels of ±5 V. The detectors, whose natural frequency is 107 kHz, were connected to the generator and the samples. The two detectors were placed on both sides of mortar samples and kept them in a straight line. Vaseline was used as the coupling agent between the detectors and samples. Then, the data and pictures were showed and saved through the connected computer. After that, the Fast Fourier Transform (FFT) of Origin software was used to transform the ultrasonic signal from time domain to frequency domain. Figure 2b shows the ultrasonic waveform and frequency test device and the typical ultrasonic waveform of mortar sample under different conditions.

### 2.4. Characterization of Cement Paste and Self-healing Products

The X-ray diffraction (XRD, Bruker (Beijing) Co., Ltd., Beijing, China) and Thermogravimetric (TG, Mettler Toledo Co., Ltd., Swiss) analysis were used to characterize the hydration products of cement pastes cured 28 days and the self-healing products produced after self-healing 30 days. The hydration products were scraped and collected from the fracture surfaces of cement pastes after cracked. The self-healing products were scraped and collected from the inner surfaces of the micro-cracks after self-healing. The samples were scanned from 10° to 80° by an X-ray diffractometer (Bruker (Beijing) Co., Ltd., Beijing, China). For TG analysis test, the samples were subjected to temperatures escalating from 25 °C to 1000 °C at a rate of 10 °C/min.

Backscattered electron (BSE) images of cement pastes were used to show how the extent of un-hydrated cement grains at 28 days and compare after self-healing 30 days. Firstly, the samples were drying under vacuum. To increase the impregnation depth of resin, pre-polished was employed on one surface of the sample using sand paper of grit size 220. The pre-polished surface was impregnated with a low-viscosity epoxy under vacuum to prevent the loose particles from falling during the next polishing. After the hardening of epoxy, the pre-polished surface was ground again using the sand paper of grit sizes 320, 800, 1200, 3000, and polished with diamond paste (6,3,1 μm) until the cement particles became visible. The images of polished surface were taken with a BSE detector. The working voltage is 20 KV. The same pair of ratios and brightness were used for all samples to ensure image grayness consistency and test repeatability.

Scanning electron microscope with an energy dispersive X-ray spectrometer (SEM-EDS, Zeiss Gruppe, Oberkochen, Germany) tests were used to characterize the microstructure and chemical composition of self-healing products. The samples were coated with gold to promote conduction and improve the image quality. The current of 50 mA and the coating time of 30 s were setting. After that, the samples were placed in a high vacuum mode to observe, immediately. The SEM-EDS were operated at an accelerating voltage of 20 keV and a working distance of 6.5 mm.

## 3. Results and Discussion

### 3.1. Self-healing Efficiency through Optical Microscope Observation and Ultrasonic Tests

#### 3.1.1. Optical Micrograph of Cracks Closure

Optical microscope observation is the most intuitive method to evaluate the self-healing of cracks, whose width was measured directly (Figure 3). Two edges of t crack before self-healing were clearly appearance. The initial crack width values, which are the average of three readings for each sample, of F1, F2, F3, F4, F5 and F6 specimens after damage are 317 μm, 275 μm, 322 μm, 288 μm, 298 μm and 311 μm, respectively. When the self-healing reached 30 days in water, the cracks were sealed and the sample surfaces were covered by the white crystalline substance, which was speculated to be CaCO_3_ precipitates [3,37]. With the increased of content of CaCO_3_ precipitates, the crack width gradually reduced.

Different samples show different self-healing effects for cracks. The crack of F3 sample was completely healed, which has a large amount of cement particles in the range of 30~60 μm. With the content of cement particles (larger than 60 μm) increased, there showed a poor crack closure such as the F6 sample. The crack of F1 sample was also not closed completely due to a lot of fine cement particles (smaller than 30 μm). The optical micrograph results reveal that the excessive coarse or fine cement particles are harmful to the self-healing of 28 days concrete. The reasons are that the hydration of cement particles less than 30 μm is fast, while the hydration only happened on the surface of coarse cement particles [38,39,40]. In other words, there was low hydration potential for F1 sample and a small amount of Ca(OH)_2_ content for F6 sample, which were cured 28 days. The smaller crack width may not be the important role in this experiment, because of the largest crack width for F3 sample with the best self-healing effect.

To numerically distinguish the extent of cracks closure, the crack width (CW) closure rate is defined as
CW closure rate = (W_0_ − W)/W_0_ × 100%(1)
where W_0_ and W are the initial crack width and crack width after self-healing, respectively. The crack width closure rate of each sample is shown in Figure 4.

Corresponding to the optical micrographs of crack closure, the F3 sample presents a largest closure rate of the crack. The closure rates of F1 and F6 samples are less than 40%. The difference in self-healing effect of each sample is significant by the comparison of the crack width closure rate.

#### 3.1.2. Ultrasonic Tests

Different conditions of concrete have different UPV values. The relatively low value of the UPV was caused by the air entrapped in the cracks and the recovery value was that because the UPV propagates through the solid phase, which produced by self-healing, instead of through the air phase [41,42]. Figure 5a shows the UPV values of un-cracked, pre-cracked and self-healing samples, respectively. Based on the significantly effect of saturation degree for the UPV results, all samples were dried in a 60 °C environment for 24 h to keep a same saturation degree. The self-healing efficiency of concrete is represented by UPV healing rate as shown in Figure 5b. From Figure 5a, the UPV values of all un-cracked samples are in the range of 3200 m/s to 3400 m/s, indicating that the ultrasonic wave travels quickly in mortar samples. The small difference of UPV values may be caused by their internal defects. The UPV values decreased to 1400 m/s because of the reflection and dispersion of ultrasonic waves when samples pre-cracked [43]. The difference of these UPV values may be caused by the various sizes and shapes of cracks. After self-healing, the UPV values increased compared with the values of cracked samples. The reason is that the cracks were filled and sealed by the hydration products and CaCO_3_, which made the cracks narrow or even close [37]. The different healing degree of cracks makes the UPV values difference. The F3 sample has the highest recovery of UPV values after self-healing, while the other samples present the poor recoveries. It reveals that there were more self-healing products to fill the crack of F3 sample while less products were to fill the cracks of F1, F5 and F6 samples.

To present the self-healing effects more clearly, the ultrasonic pulse velocity (UPV) healing rate is defined as
UPV healing rate = [1 − (V_1_ − V_3_)/(V_1_ − V_2_)] × 100%(2)
where V_1_, V_2_ and V_3_ indicate the UPV of mortars which is un-cracked, pre-cracked and self-healing, respectively. The UPV healing rates are shown in Figure 5b.

It can also be seen from Figure 5b that the UPV healing rate of F3 sample is higher than other samples. The trend of UPV healing rate is consistent with the trend of crack width closure rate. However, the UPV healing rates of all samples are less than 100%, which indicates that the density of the crack after self-healing is lower than that of the matrix. One of the reasons is that there are unhealed regions in the interior cracks [44]. UPV values can reflect the self-healing inside the crack more realistic than the optical micrographs for crack.

Ultrasonic waveforms and frequencies carried ultrasonic waves transmitted through different media (paste, aggregates, pores and cracks). Therefore, the self-healing properties of concrete can be evaluated by detecting these parameters [34]. Ultrasonic waveform also shows high repeatability due to the energy transmission in a stable state. Figure 6 shows the ultrasonic waveforms of mortar samples and Figure 7 shows the ultrasonic frequencies of samples.

The ultrasonic waveforms of un-cracked samples are significant and present that the maximum amplitudes among all peaks are larger than 20 mV as shown in Figure 6. When the samples cracked, the maximum amplitudes decreased obviously. The result is due to that the ultrasonic energy was weakened by the strong interference of the air in cracks [45,46]. Fortunately, the maximum amplitudes of samples increased when cracks self-healed. The maximum amplitude of F3 sample, especially, increased significantly and showed the best self-healing effect. The reason may be that there were more cement particles in the range of 30~60 μm than other samples, which had strong continued hydration potential for 28 days concrete and produced much hydration products. However, the ultrasonic maximum amplitudes of self-healed samples are still lower than that of the un-cracked samples, which are consistent with the results of UPV test.

Interference of noise signals in the ultrasonic waveform is inevitable, especially the cracked samples. To show these changes accurately, the Fast Fourier Transform (FFT) with a Gaussian function was used to get more pronounced and clear results [47]. The ultrasonic signal of time domain can be transformed to the signal of frequency domain by FFT, which can also evaluate the variation of ultrasonic waves [48]. Figure 7 shows the ultrasonic frequency graphs obtained by FFT. The dominant frequencies of un-cracked samples are at a resonance frequency of 107 kHz. However, the peaks appearing in the low frequency range are mainly caused by the pores in the mortars [47]. When the samples cracked and the ultrasonic energy was weakened by the cracks, it is hardly to find the existence of the dominant frequencies. However, the dominant frequency of F3 group recovered significantly after self-healing, which indicates that it has the best self-healing effect. It is also consistent with the results of crack width closure and UPV test.

It may be more intuitive to express by the recovery rate of amplitude and frequency. The ultrasonic shape (US) healing rate is defined as
US healing rate = [1 − (S_1_ − S_3_)/(S_1_ − S_2_)] × 100%(3)
where S_1_, S_2_ and S_3_ indicate the maximum amplitude of ultrasonic wave for un-cracked mortars, cracked mortars and self-healing mortars, respectively. And the ultrasonic frequency (UF) healing rate is defined as
UF healing rate = [1 − (F_1_ − F_3_)/(F_1_ − F_2_)] × 100%(4)
where F_1_, F_2_ and F_3_ indicate the frequency amplitude of ultrasonic wave for un-cracked mortars, cracked mortars and self-healing mortars, respectively. The US and UF healing rates are shown in Figure 8.

The intuitive results, as shown in Figure 8, present that the self-healing efficiency of F3 sample is the best. Considering the CW closure rate and UPV healing rate, it can be determined that the F3 sample has good self-healing ability of concrete at 28 d.

Based on the above results, the ultrasonic nondestructive tests can reflect the self-healing of internal cracks, compared with the optical micrographs of the cracks closure. Although the crack of F3 sample is completely closed, the internal filling is not as dense as the cement matrix, which was proved by the recoveries of ultrasonic pulse velocity, waveform and frequency. The difference of self-healing ability may be caused by the PSD of cement and the detailed analysis will be below.

#### 3.1.3. The Self-healing Ability Related to the PSD of Cement 

To explore the influence of PSD on the self-healing ability, the corresponding relationship between the volume percentage of cement particles in different size ranges and the self-healing rates are established, as shown in Figure 9.

It can be seen that the trend of the percentage of cement particles in the range of 30~60 μm is consistent with that of the self-healing rate changes. It can be speculated that the cement particles sizes in this range have a great influence on the self-healing ability of concrete cured 28 days and it may be related to their hydration degree. In comparison, the cement particles, whose size is less than 30 μm, perform a high degree of hydration when the samples cured 28 days. However, the hydration only happens on the particles surfaces for the coarse cement. Even if there is sufficient water in the crack, therefore, the continued hydration of cement for these fine and coarse particles is weak. The results are caused by that there is less un-hydrated cement for F1 sample and it is difficult to continued hydration for F6 sample because of the covering and blocking of hydration products on cement particles surface. The cement particles in the range of 30~60 μm still have a great hydration potential on concrete cured 28 days and they also could hydrate to produce more Ca(OH)_2_, which is beneficial to the precipitation of CaCO_3_. This is the reason of that the F3 sample has the best self-healing ability, which is consistent with the above experimental results.

The difference of the self-healing ability is mainly due to the different amount of self-healing products. So, it is necessary to characterize and analyze the hydration degree of cement pastes before healed and the self-healing products.

### 3.2. TG Analysis of Cement Paste and Self-healing Products

To determine the hydration degree of cement paste, TG analysis were used to evaluate the relative content of Ca(OH)_2_. Figure 10 shows the TG curves and the derivative thermogravimetry (DTG) curves of cement pastes before self-healing.

The F1 sample has a large amount of Ca(OH)_2_ and non-evaporating water as shown in Figure 10. It is caused by the high degree of cement hydration. The Ca(OH)_2_ contents of the F2 and F3 samples are slightly reduced. In the F5 and F6 samples, the Ca(OH)_2_ contents and the weight losses of non-evaporating water are significantly reduced, which demonstrate that these samples have low degrees of cement hydration. In other words, there is much un-hydrated cement in F5 and F6 samples, which is beneficial to hydrate continuously. However, the results show that the F3 sample has the best self-healing ability. It needs to be explained by analyzing the self-healing products.

TG analysis was also used to characterize the self-healing products of F2 and F3 samples because of the most products production (Figure 11). It shows the TG curves have the significant weight loss and the DTG curves have the distinct endothermic peaks in the range of 70 °C to 200 °C, which may be caused by the decomposition of Ettringite or the dehydration of the C-S-H gel [49]. It indicates that the un-hydrated cement continues to hydrate. The samples show weight loss again by the TG curves decline from 400 °C to 720 °C, but there are no obvious endothermic peaks in the DTG curves. It may be the decomposition of un-carbonized Ca(OH)_2_ and the amorphous calcium carbonate with a very small amount. However, it is difficult to distinguish between Ca(OH)_2_ and amorphous CaCO_3_ by TG and DTG method [50]. Subsequently, the TG curves decrease sharply and the DTG curves show significant endothermic peaks from 720 °C to 840 °C. This is mainly because of the decomposition of massive calcite [51]. Overall, the results show that calcite is the most important self-healing product, while there is also some continued hydration product.

Based on the TG and DTG analysis results of cement pastes and self-healing products, a large amount of Ca(OH)_2_ produced by early hydration of the F1 sample was wrapped in the matrix and was difficult to form CaCO_3_, which caused the poor self-healing ability. In contrast, the F6 sample has a lot of cement particles larger than 60 μm, which only hydrate on the surface of cement particles. Therefore, the low content of Ca(OH)_2_ and the difficulty of continued hydration cause the poor self-healing ability. However, the F3 sample, which has some un-hydrated cement particles and a certain amount of Ca(OH)_2_, show the best self-healing ability through the above test results. This is caused by a co-action of CaCO_3_ precipitation and continued hydration. Furthermore, the un-hydrated cement is more easily to release Ca^2+^ when there is sufficient water and it is beneficial for the precipitation of CaCO_3_. Actually, the cement particles with size of 30~60 μm promote this effect.

### 3.3. XRD Analysis of Cement Paste and Self-healing Products

XRD analysis can also determine the products types of hydration and autogenous self-healing. Their peak intensities represent the relative content of products. Thus, the XRD analysis of cement paste before self-healing was used to compare the relative contents of Ca(OH)_2_ and un-hydrated cement, as shown in Figure 12.

As shown in Figure 12, the Portlandite peaks are significantly higher than the peaks for calcium silicate of F1 sample. It indicates that there are much content of Portlandite and relatively less content of un-hydrated cement. This is because the F1 sample has many fine cement particles which hydrate quickly. With the increasing content of coarse cement particles, the Portlandite peaks are gradually reduced and the peaks of un-hydrated C_3_S and C_2_S become high relatively, which represents a decline in the degree of hydration. The contents of Portlandite and un-hydrated cement in F3 sample are moderate and its cement particle sizes especially have great potential for continued hydration. Although there are a large amount of un-hydrated cement in F6 sample, the self-healing ability is really poor because that the volume percentage of cement particles with size of 60~150 μm is comparatively large, which hydrates only on the surface of cement particles and the continued hydration of cement nuclei is difficult because of the surface covering.

The XRD analysis also confirms that CaCO_3_ is the mainly component of the self-healing products, as shown in Figure 13. The peaks of Ca(OH)_2_ are hardly found, which indicates that the Ca(OH)_2_ in the crack surfaces reacted with dissolved in water [11,50,52]. However, the Portlandite produced by the early hydration of F1 sample was wrapped by C-S-H gel. Consequently, the Portlandite is difficult to dissolve and form CaCO_3_, which reduces the self-healing ability. In contrast, the F3 sample contains a certain amount of Portlandite which is beneficial to the formation of CaCO_3_. The Ca^2+^ released by the continued hydration could accelerate the CaCO_3_ crystals growing, therefore, the continued hydration promoted the development of carbonation, which improved the self-healing of concrete. However, the CaCO_3_ produced by the self-healing has different mineral phases. The major product is calcite because of the highest peak intensity, while the weak peaks of aragonite and vaterite are also found. The different mineral phases may be related to the water environment and the content of CO_2_ [53]. Although the XRD results show that the self-healing product is mainly calcite, it only indicates that carbonation reaction played an important role in the autogenous self-healing. However, the effect of continued hydration of un-hydrated cement cannot be determined, which needs to be explored by other testing methods.

### 3.4. BSE Image Analysis of Cement Paste

BSE images of cement pastes cured 28 days and self-healing 30 days were carried out, as shown in Figure 14. The gray level in the BSE images corresponds to un-hydrated cement, hydration products and pores from light to dark. The white spots, which were circled by red dotted line, represent the un-hydrated cement particles of the matrix in BSE images [51]. For the samples cured 28 days, F5 group has many un-hydrated cement particles, but most of them are larger than 90 μm. The least content of un-hydrated cement was found in F1 group. Conclusions can be made that the degree of hydration decreases with the cement particles get coarser, which is in agreement with the previous study. When the samples self-healed 30 days, the degree of hydration is further improved, especially the area near the crack surfaces. It is speculated that the continued hydration promote the autogenous self-healing of concrete cracks.

### 3.5. SEM and EDS Analysis of Self-healing Products

There are some continued hydration products found by TG analysis to heal the cracks. Further, the SEM analysis could also present the continued hydration products of F3 samples, which produced in the early autogenous self-healing process, as shown in Figure 15a. The major autogenous self-healing product which produced in the later self-healing process is CaCO_3_, as shown in Figure 15b.

A large amount of needle-like products were found on the crack surface after healed 7 days and they were interlaced into a loose network structure as shown in Figure 15a. These may be the new hydration products (Ettringite or C-S-H gel). Actually, they are distinguished from the hydration products in the cement matrix through the morphology and compactness. In addition, it also reveals that the hydration products and CaCO_3_ coexisted. The agglomerates in the network may be the CaCO_3_. Therefore, the SEM result could confirm that there were new hydration products in the initial period of self-healing, accompanied by the CaCO_3_ precipitation. 

The microstructures of the self-healing products after healed 30 days were also observed by SEM to confirm that they are different from the early self-healing products, as shown in Figure 15b. There is no loose network structure, replaced by the CaCO_3_ covered on the internal crack surface. Thus, it is proved that the precipitation of CaCO_3_ was mainly self-healing product at a later stage of self-healing. Moreover, the CaCO_3_ mainly exists in the form of calcite with different shapes, such as the block shape and the parallel plate. The formation of different shapes is perhaps caused by the different crystalline sites and crystal growth orientation. However, the calcite also brings some large pores because of the different shapes, leading to that the self-healing rates cannot achieve 100% based on the ultrasonic tests.

The SEM results show that the early self-healing products are produced by the continued hydration and carbonization, while the later self-healing products are mainly composed of CaCO_3_. Furthermore, the SEM-EDS analysis on the cross section of healed cracks was performed to reveal that the continued hydration products were covered by the calcite and the continued hydration products were also quantitatively characterized. The selection of the target area is shown in Figure 16a and the linear scan and EDS mapping for the target area are shown in Figure 16b. The atomic ratio plot of the self-healing products is shown in Figure 17 to illustrate the characteristics of the healing products observed.

The SEM-EDS results show that the self-healing products are mainly composed of O, Si, C, Al and Ca, as shown in Figure 16. The difference in the content of various elements at different positions indicates the difference of self-healing products. According to the EDS results, there are a large amount of hydration products closest to the hardened paste and formed a product layer with a thickness of 12~22 μm, which suggests that the continued hydration occurred on the surface of cracks at the early age of autogenous self-healing. However, the Ca/Si ratio in the new hydration product is lower than that in the hardened paste. Some calcium ions may be released into the water and form the calcium carbonate. In the later stage of autogenous self-healing, a large amount of CaCO_3_ adhered to the surface of the continued hydration products and filled the cracks. Therefore, it can be shown that the self-healing was carried out under the combined action of continued hydration and carbonation. Figure 17 shows the continued hydration products and the carbonization products. When the composition of calcium is overwhelming compared to silicon, it indicates that CaCO_3_ is produced as the calcium rich crystals [54]. While the gel-like materials are the continued hydration products because of a certain amount of silicon and aluminum. Actually, Calcite precipitation is the main factor, which is proved by the XRD and TG analysis.

## 4. Conclusions

Different particle size distribution of cement has different effect on concrete self-healing. When the sample has the optimized cement particle size, it shows the best self-healing ability of concrete. The ultrasonic non-destructive tests can reflect the self-healing of internal cracks more virtually than the surface cracks closure test. The conclusions are as follows:

1. For the Portland cement containing more particles with the size of 30~60 μm, the surface crack closure is more complete and the recoveries of ultrasonic signal are better. Therefore, the concrete could achieve a better self-healing ability of concrete at 28 days.

2. For the methods to characterize the self-healing ability of concrete, the ultrasonic tests are more accurate in characterizing the self-healing of internal crack than optical micrograph observation, especially the ultrasonic waveform and frequency tests.

3. The autogenous self-healing of concrete is jointly affected by the continued hydration and carbonation. The autogenous self-healing of concrete is mainly controlled by the continued hydration at the early period of self-healing, while the carbonation plays an important role at the late stage of self-healing. Calcite is the mainly form of self-healing products and plays a role in the whole self-healing process, while the continued hydration products interlace into a loose network structure in the early stages of self-healing.

4. The cement particle size could affect the continued hydration by affecting un-hydrated cement content and the carbonation by affecting the Ca(OH)_2_ content. Therefore, a proper distribution of cement particle size, which brings a suitable amount of Ca(OH)_2_ and un-hydrated cement, could improve the self-healing ability of concrete.

## Figures and Tables

**Figure 1 materials-12-02818-f001:**
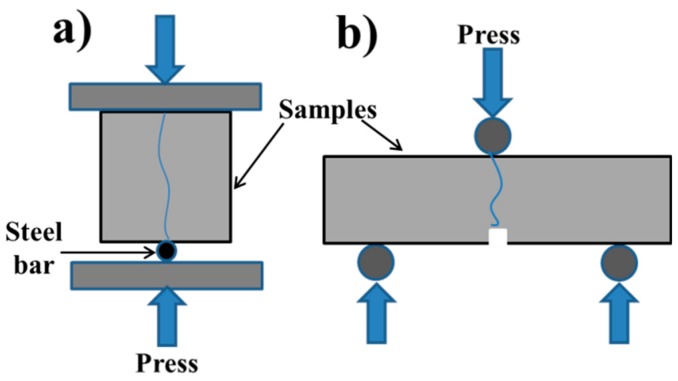
The schematic of prefabricating micro-cracks. (**a**): the schematic of the splitting test; (**b**): the schematic of the three-point bending test.

**Figure 2 materials-12-02818-f002:**
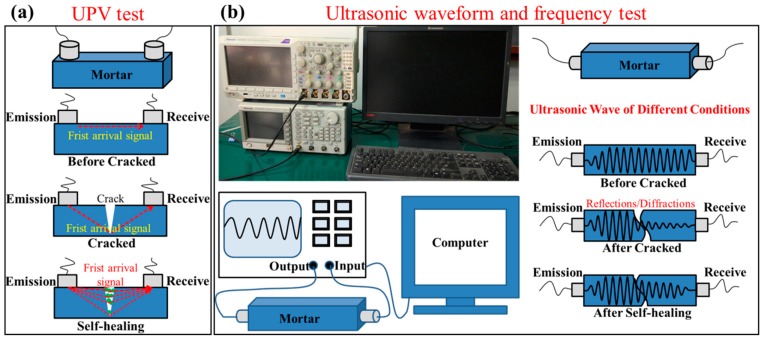
Ultrasonic test device and schematic diagram. (**a**): the schematic diagram of the UPV test, (**b**): the ultrasonic waveform and frequency test device and the typical ultrasonic waveform.

**Figure 3 materials-12-02818-f003:**
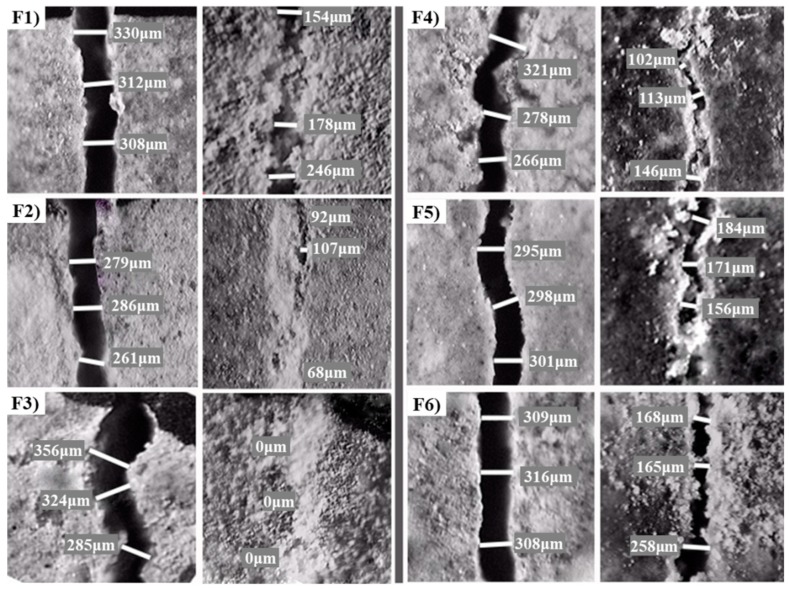
Optical micrographs of cracks closure (The left pictures show the crack before self-healing and the right ones show them after self-healing for each group; (**F1**–**F6**) are the group numbers of cement particles with the average size 22.82 μm, 29.78 μm, 39.10 μm 49.96 μm, 60.51 μm and 73.60 μm, respectively.)

**Figure 4 materials-12-02818-f004:**
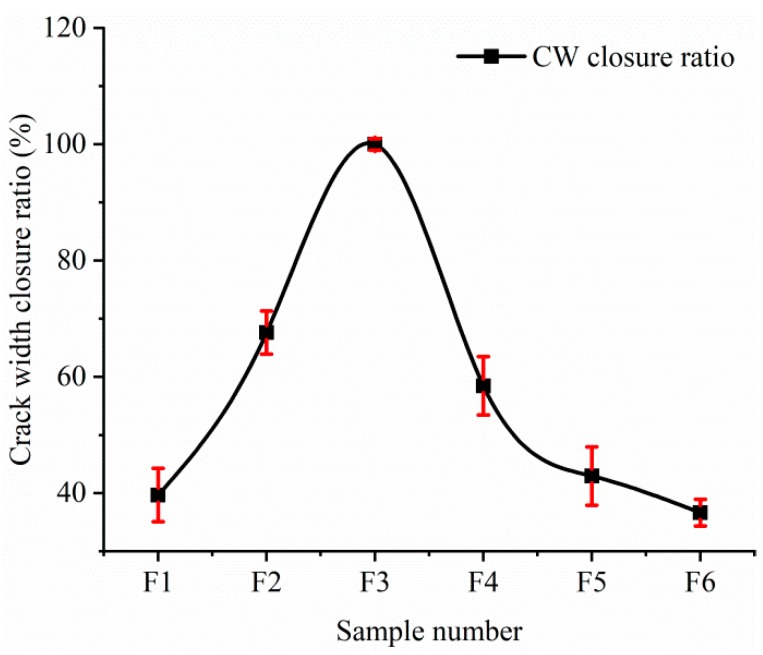
Crack width closure rate.

**Figure 5 materials-12-02818-f005:**
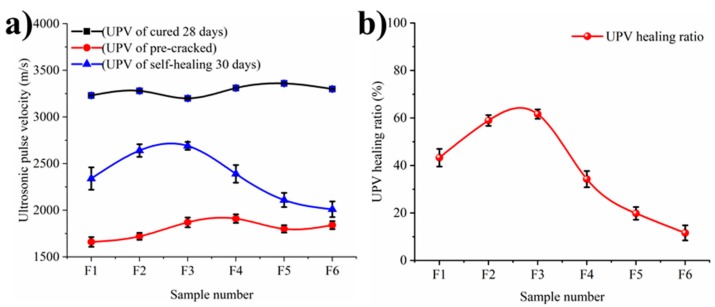
Ultrasonic pulse velocity of samples cured 28 days. (**a**): the UPV values of un-cracked, pre-cracked and self-healing samples, respectively; (**b**): the UPV healing rates.

**Figure 6 materials-12-02818-f006:**
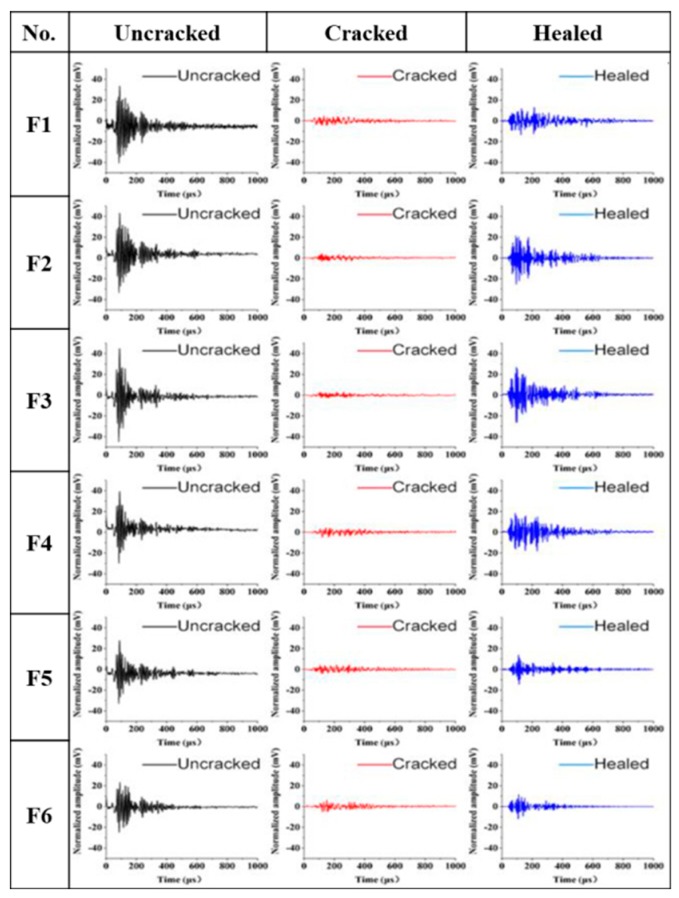
Ultrasonic waveform analysis of samples. (**F1**–**F6**) are the group numbers of cement particles with the average size 22.82 μm, 29.78 μm, 39.10 μm 49.96 μm, 60.51 μm and 73.60 μm, respectively.

**Figure 7 materials-12-02818-f007:**
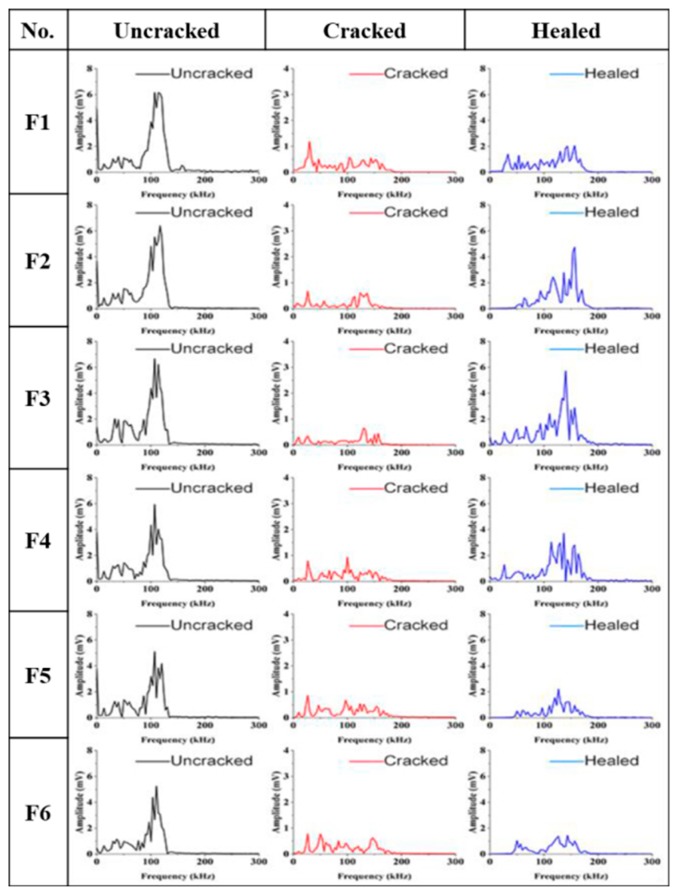
Ultrasonic frequency analysis of samples. (**F1**–**F6**) are the group numbers of cement particles with the average size 22.82 μm, 29.78 μm, 39.10 μm 49.96 μm, 60.51 μm and 73.60 μm, respectively.

**Figure 8 materials-12-02818-f008:**
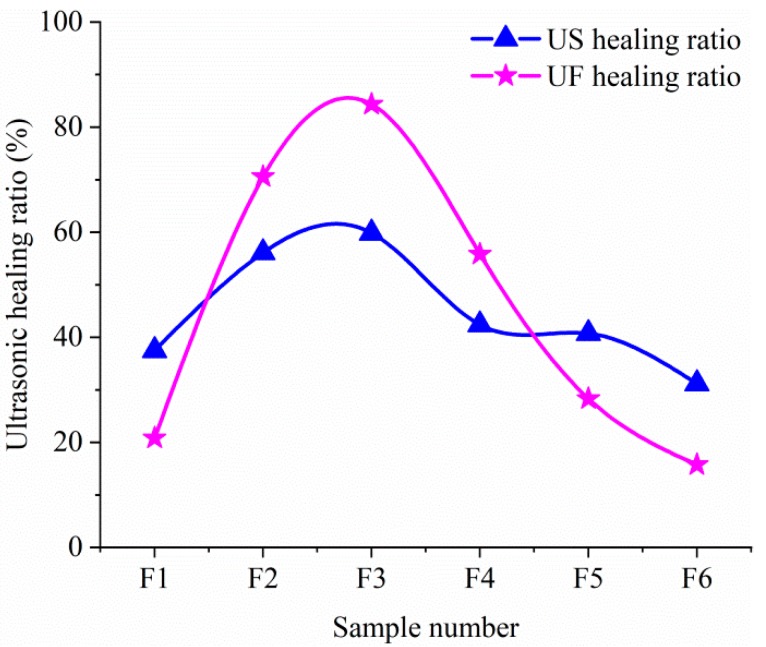
The US and UF healing rates of samples.

**Figure 9 materials-12-02818-f009:**
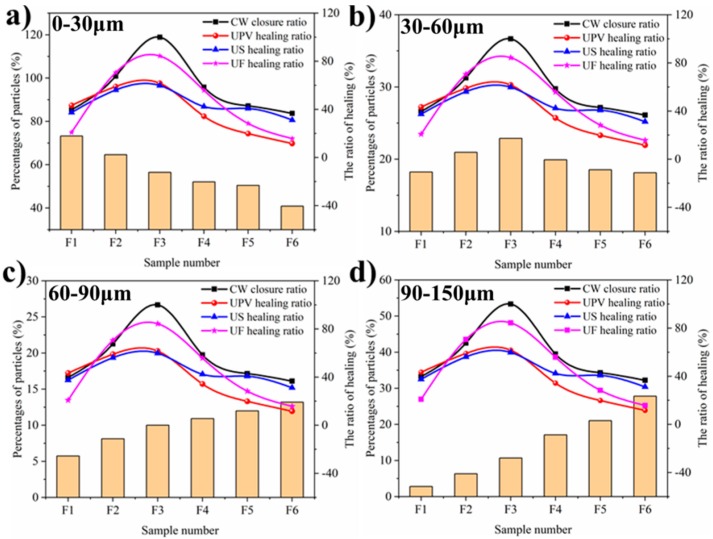
The relationship of self-healing rates with cement particles sizes ((**a**–**d**) indicate the volume percentages of cement particles within a range of less than 30 μm, 30~60 μm, 60~90 μm, and more than 90 μm, respectively).

**Figure 10 materials-12-02818-f010:**
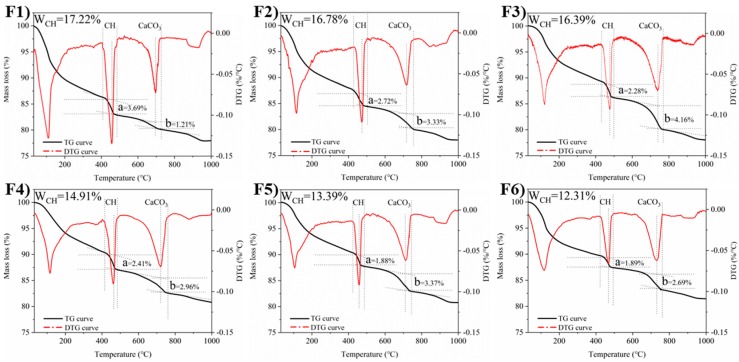
TG analysis of cement pastes before self-healing. (**F1**–**F6**) are the group numbers of cement particles with the average size 22.82 μm, 29.78 μm, 39.10 μm 49.96 μm, 60.51 μm and 73.60 μm, respectively.

**Figure 11 materials-12-02818-f011:**
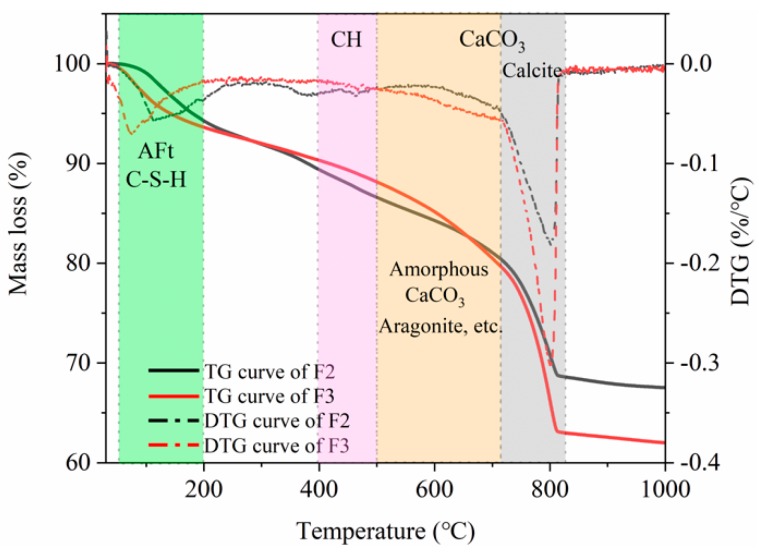
TG analysis of self-healing products.

**Figure 12 materials-12-02818-f012:**
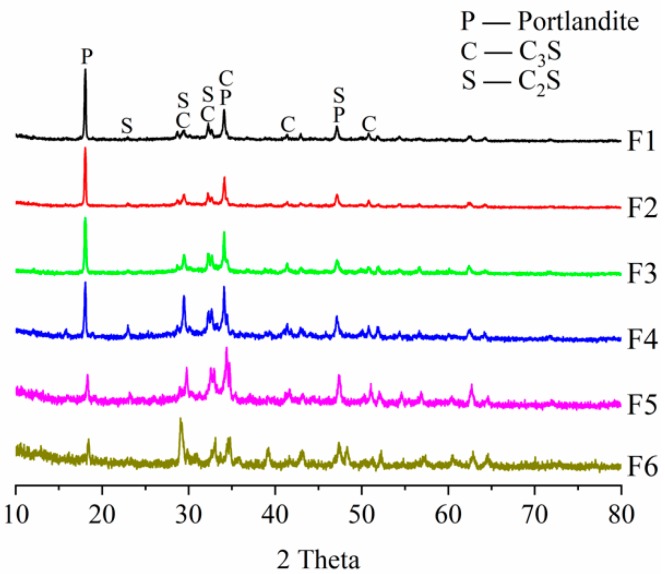
XRD analysis for the cement pastes before self-healing.

**Figure 13 materials-12-02818-f013:**
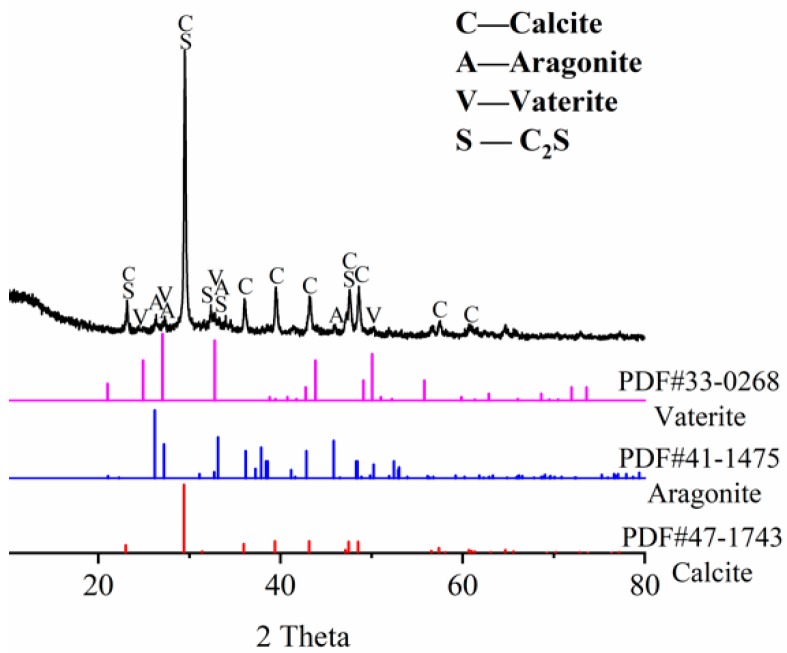
XRD analysis of self-healing products.

**Figure 14 materials-12-02818-f014:**
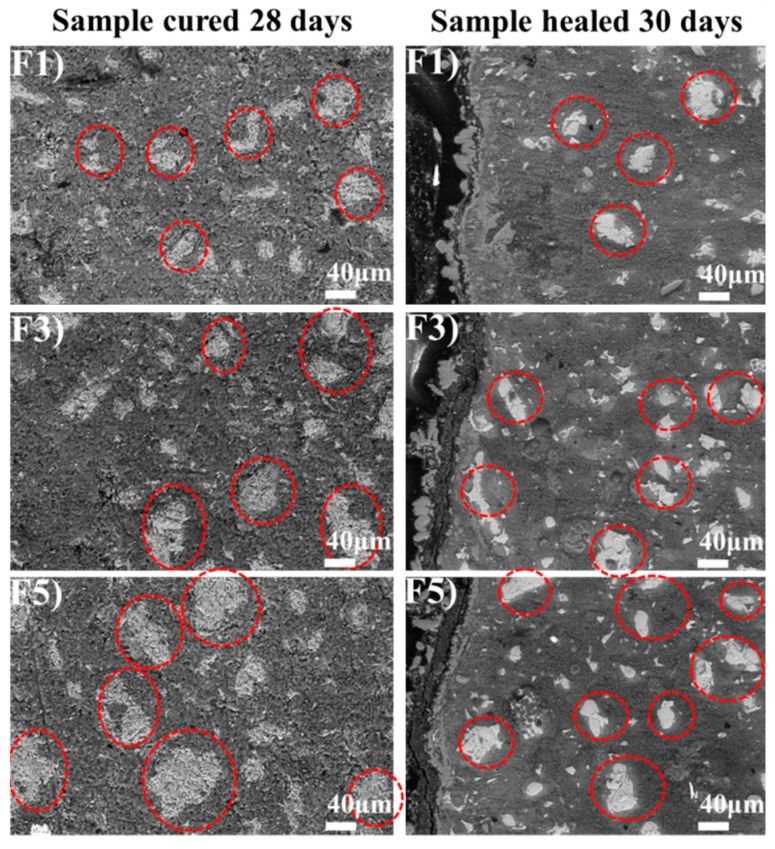
BSE image analysis of cement paste. (**F1**,**F3**,**F5**) are the group numbers of cement particles with the average size of 22.82 μm, 39.10 μm and 60.51 μm, respectively.

**Figure 15 materials-12-02818-f015:**
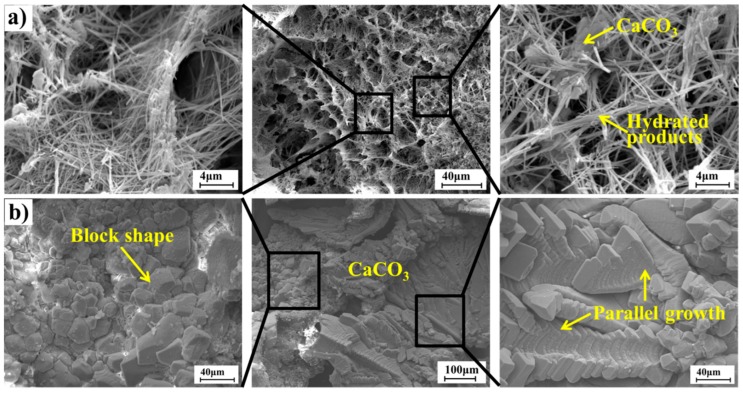
SEM analysis of self-healing products (**a**) is self-healing for 7 days; (**b**) is self-healing for 30 days.

**Figure 16 materials-12-02818-f016:**
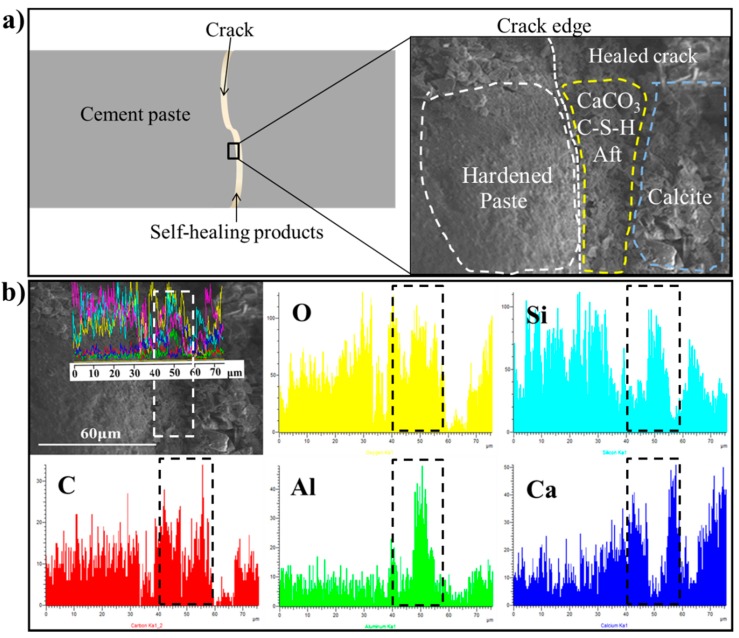
SEM-EDS analysis of cross-section after self-healing for 30 days. (**a**) is the target area of SEM-EDS; (**b**) is the linear scan and element distributions for the target area.

**Figure 17 materials-12-02818-f017:**
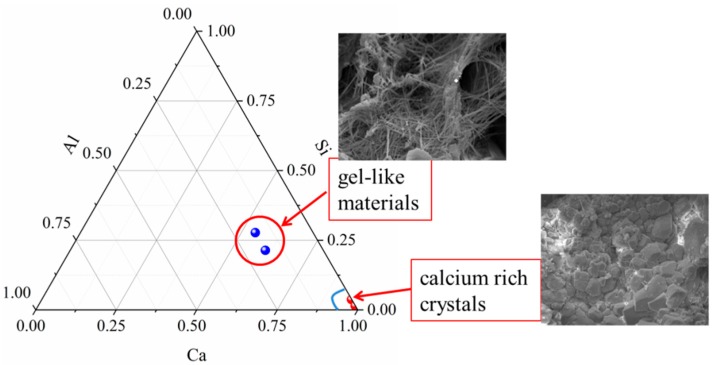
Atomic ratio plot of the self-healing products.

**Table 1 materials-12-02818-t001:** Chemical compositions of cementitious materials.

Chemical Components	Clinker (%)	Gypsum (%)
CaO	63.55	41.70
SiO_2_	22.88	2.01
Al_2_O_3_	4.19	0.52
MgO	3.97	0.80
Fe_2_O_3_	3.30	0.39
K_2_O	0.64	-
Na_2_O	0.52	-
SO_3_	0.43	41.78
Loss on ignition	0.52	1.12
Crystal water	-	11.68

**Table 2 materials-12-02818-t002:** Volume percentage of cement particle size distribution.

No.	Mean Size (μm)	Volume Percentage (%)				
		<10 μm	10~30 μm	30~60 μm	60~90 μm	>90 μm
OPC	14.29	47.21	38.53	13.71	0.55	0
F1	22.82	38.27	35.00	18.22	5.73	2.78
F2	29.78	34.86	29.76	20.95	8.12	6.31
F3	39.10	29.26	27.17	22.88	10.00	10.69
F4	49.96	27.36	24.72	19.91	10.92	17.09
F5	60.51	24.97	23.49	18.53	11.98	21.03
F6	73.60	20.61	20.26	18.13	13.19	27.81

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
