# Peer review of "Research on the Improvement of Concrete Autogenous Self-healing Based on the Regulation of Cement Particle Size Distribution (PSD)"

_materials, 2019, doi:10.3390/ma12172818_

Round 1
Reviewer 1 Report
I offer the following comments to the authors:
-This study is focused on the autogenous healing phenomenon in concrete. I believe the keyword 'autogenous' should be referenced, as there is no mention of it in this text.
-On line 71, background section, a citation would be beneficial citing the link between fine cement particles and poor durability in concrete.
-Much of the background section gives insight on other self-healing processes, but has very few citations on the influence of portland cement's particle size on the autogenous healing of concrete or mortar.
-On section 2.2.1, do any of these sample groups resemble the particle size distributions of commercially available cements (e.g. Type I or Type III portland cement)?
-On section 2.2.2, there should be mention of how much cementitious content was used in the mixture design (in kg/m3)
-On section 2.2.3, how was the cracking controlled? Was a displacement-controlled loading rate used? If so, at what rate? It is also unclear why the samples were filled with epoxy resin after cracking. Lastly, this section should also mention the period of time in which the samples were left to heal.
-On section 2.4, there should be mention of the accelerating voltage used by the SEM-EDS, and provide more details on sample preparation (e.g. was the sample placed in low vacuum mode, etc.) and the time in which they were viewed. While this was later mentioned in the text, the methodology section should include these details.
-In section 3.1, there is no mention of the initial crack widths for all specimens right after damage. This is a very important data set that is missing, as there is no way to tell if a certain sample group healed faster due to the enhanced autogenous healing of concrete (attributed to a cement particle size distribution), or because the crack sizes were smaller for one group than the other.
-In section 3.1.2, what was the condition of the mortar samples when testing the UPV at the uncracked, pre-cracked, and self-healing stages? The degree of saturation of concrete can significantly affect the UPV results.
-The discussion on hydration kinetics vs. particle size could use citations to verify findings. In addition, SEM images of freshly cracked concrete surfaces would be beneficial to see extent of unhydrated cement particles exposed (and thus the potential for further hydration) for each sample group.
-In section 3.4, an atomic ratio plot would be beneficial to illustrate the characteristics of the helaing products observed, and give an indication on the extent of newly formed calcium rich crystals vs. gel-like materials.
Reviewer 2 Report
Research on the improvement of concrete intrinsic self-healing based on the regulation of cement particle size distribution
Comments from Reviewer
The work claims at investigating the influence of cement particle size distribution on self-healing capacities of concrete, using non-destructive ultrasonic testing and optical micrography to evaluate the self-healing performance and find the optimal particle size.
Comments:
The work is interesting and the application promising. However, the manuscript should be revised to guarantee its publication.
1. Abstract:, lines 22-23, the following sentence has a lack of meaning/significance. Please revise:
“The CaCO3 precipitation is the major factor for self-healing, while the continued hydration is affected by the particle size of cement and the content of unhydrated cement.”
2. Introduction:
In line 35, the expression “a lot of” cannot considered as proper academic writing. Please revise.
In line 60, the following sentence seems confusing. Please revise.
“Therefore, repair agent with low-cost has been recently composed to be an advanced and good technique to develop concrete self-healing.”
In line 80, the XRD, TGA and SEM/EDX techniques should be included to indicate which experimental methods have been used to evaluate the contribution of continued hydration.
Sufficient detail and a rigorous description of the materials properties is perceived.
3. Experimental:
a. In the Materials journal, this section should be placed after the results and discussion is presented.
b. Table 1 does not really show the complete percentage of the clinker nor the gypsum used in the research. The sum of the chemical components should be 100%, when it is not. Please revise or indicate which components are missing.
4. Results:
a. The number of samples prepared and tests performed are intensive and gives rise to sufficient statistics to draw conclusions.
b. It is not clear in Figures 14 and 15 which particle size are evaluating.
5. The conclusions are clear and properly supported by the experimental results. I only recommend the authors to draw an ending conclusion from the results in a more analytical way, without using the numerical results obtained in the experiments
As I said, the work is interesting and well conducted. Some minor typing errors should be corrected.
Reviewer 3 Report
General remarks:
The topic proposed in the paper is interesting, original and within the scope of the Journal. The approach followed is quite well documented and clearly illustrated in the text.
For these reasons, I recommend publication in Materials
The paper reports interesting results; before accepted for publication, major revisions are necessary by considering the following issues:
Major points:
Introduction:
Generally, there is a lack of self-healing terminology.
As the authors know, the JCI (Japanese Concrete Institute) TC-075B first provided definitions of self-healing which were further taken as reference by RILEM (Reunion Internationale des Laboratoires et Experts des Materiaux, Systemes de Construction et Ouvrages) Ex221-SHC.
Please, consider the possibility to first present the different mechanisms of self-sealing/healing properly and then discuss and critically review the different self-healing engineering techniques.
Finally, taking into account the aim of the research, which is to underline the effect of the cement particle size, I would suggest improving the introduction especially in relation to the autogenous self-healing as well as the autonomic healing.
Pag 2_line43: Could the author add some proper references?
Moreover, the main method herein presented to evaluate the healing is the UPV measurements. Could the authors add also here the references, highlighting the main results and conclusion?
Materials and Methods:
Materials:
Could the authors indicate the class of the used cement, according to the reference Standard?
Could the authors also specify the type of sand and the particle size?
Samples preparation:
Could the authors indicate the Standards taken as reference for the tests as well as for the samples size?
Could the authors better described the casting procedure (mould type, time before de-moulding, etc).
Pre-cracking phase:
Could the authors indicate proper references for the pre-cracking phase [of both methods: fig1 (a), fig 1(b)].
Moreover, are the samples prepared for inducing the crack? Are they notched (depth of notching?)
Could the authors describe the whole set up of the test?
Could the authors clarify the aim of using epoxy resin after the pre-cracking?
What is the crack width??
UPV:
Could the authors better explain the position of the transducers? Could the author clarify why the distance between the two sensors is 97 mm, if the length is 160 mm?
Characterization of the self-healing products:
Could the authors also here specify where they collected the fragments for the x-ray diffraction?
Result and Discussion:
Assuming that one sample for each type of observed conditions is not enough to evaluate the phenomena, the optical microscopy observations seem to address some conclusions.
Firstly it seems, unfortunately only in this section of the paper, that the crack width herein considered is in the range 250-350mm. Could the authors explain why they chose this range of case study? Could also deeply explain the relationship between the crack width (that can be healed) and the cement particles size? References about the hydration process related to the cement particles are needed.
Moreover, it seems that 300 mm is the limit of the crack width that can possibly be healed. Is there any possible explanation or comments about?
Ultrasonic tests:
Could the authors add a proper reference to the methodology adopted to evaluate the UPV healing rate index? It seems different from the method used in the indicated reference.
Fig 5 (b)comment: contrarily to what the authors said, the sample with the higher UPV healing rate is the sample named F2. And the trend of the UPV rate ( fig 5(b) is clearly NOT consistent with the trend of crack (fig 4). Could the authors make comments on?
Pag7_line 199: Do the authors consider the possibility to do flexural tests after the healing stage? The flexural strength is more confidently evaluable as healing index than the UPV measurements as well as the microscope observations.
For this reason, could the authors argue why they present the studied phenomena as self-healing instead of self-sealing process?
Minor points:
Fig 3: the yellow-colored letters are not so easily readable.
Ultrasonic waveforms and frequencies:
In order to make better readable the graph in Fig 7, I would suggest having all the data on the same scale. Please, consider plotting the cracked graphs accordingly with the others (y axis).
Round 2
Reviewer 1 Report
Thank you for addressing the comments and providing a revised manuscript.
One more consideration that could add value to this study is an investigation of the microstructure of concrete or mortar produced from these five cement grain particle distributions. This can be achieved by cutting and polishing a section, and then viewing it under the backscattered image mode. In this case, the samples could be extracted to observe the concrete/mortar at 28 days of age, and then after healing. While more samples would need to be cast for that matter, it could provide more evidence to your research paper as it would be able to show how the extent of unhydrated cement grains at 28 days and compare after healing. This is of course, a bonus, as I believe this paper should be accepted with its current results.
Reviewer 3 Report
The last version of the paper “Research on the improvement of concrete autogenous self-healing based on the regulation of cement particle size distribution” has been improved. Notwithstanding, it is in the opinion of this reviewer that some clarifications will improve the quality of the research work.
Section 2.3.2. Please, specify that the Ultrasonic Pulse velocity tests were performed on each specimen in indirect (surface) transmission. In this view the scheme in Fig 2 is not correct and might confuse the readers. Please amend the scheme representing the proper method.
Section 3.1.1_line192: the reported initial crack width value is the average of three readings.
Fig. 4 and 5: Data results reported should contain the index of dispersion. Please, consider adding the error bars to the graphs.
